# The halogen bond with isocyano carbon reduces isocyanide odor

Alexander S. Mikherdov [1]✉, Alexander S. Novikov [1], Vadim P. Boyarskiy[1] & Vadim Yu. Kukushkin [1]✉

Predominantly, carbon atoms of various species function as acceptors of noncovalent interactions when they are part of a π-system. Here, we report on the discovery of a halogen bond involving the isocyano carbon lone pair. The co-crystallization or mechanochemical liquid-assisted grinding of model mesityl isocyanide with four iodoperfluorobenezenes leads to a series of halogen-bonded adducts with isocyanides. The obtained adducts were characterized by single-crystal and powder X-ray diffraction, solid-state IR and $^{13}$C NMR spectroscopies, and also by thermogravimetric analysis. The formation of the halogen bond with the isocyano group leads to a strong reduction of the isocyanide odor (3- to 46-fold gas phase concentration decrease). This manipulation makes isocyanides more suitable for laboratory storage and usage while preserving their reactivity, which is found to be similar between the adducts and the parent isocyanide in some common transformations, such as ligation to metal centers and the multi-component Ugi reaction.

[1] Saint Petersburg State University, Universitetskaya Nab., 7/9, Saint Petersburg, Russian Federation. ✉email: a.mikherdov@spbu.ru; v.kukushkin@spbu.ru

The halogen bond (XB)[1] is among the cutting edge subfields of crystal engineering because of broad application of XB in supramolecular chemistry[2], XB-involving organocatalysis[3], synthetic coordination chemistry[4], polymer chemistry[5], drug discovery[6], and, eventually, due to the role of XB in human function[7]. XB is usually defined as a noncovalent attractive interaction between an electrophilic region (σ-hole; σh) on a XB donating halogen atom and a Lewis base functioning as XB acceptor[1]. XB acceptor centers conventionally include electronegative hetero-atoms bearing lone pair(s) with many elements, such as halogens[8], chalcogens[9], pnictogens[10] etc., electron-donating π-systems, and even electron-donating positively charged metal centers[11,12]. Carbon—the structural core of all organic matter, the element that gives the highest diversity of compounds, their forms, and covalent bonds—is involved in noncovalent interactions as XB acceptor mostly as part of a π-system (e.g., double-bonded[13], triple-bonded[14], or aromatic systems[15]). The solid-state interaction involving the carbon lone pair (lp) and halogen atom has been previously observed in only one case, namely in the adduct (IAd)•IPFB formed upon the interplay of iodopentafluorobenzene (IPFB) and N-heterocyclic carbene (IAd)[16]. However, in (IAd)•IPFB, the formed C–I bond ($d_{XB}$ = 2.754(3) Å; 75% of the sum of Bondi vdW radii) exhibits a strong covalent contribution and significant charge transfer (as follows from our DFT calculations, see Supplementary Discussion). Accordingly, this linkage could be attributed to the strong XBs[17] and by nature it is closer to the so-called coordinative XB[18] as in complexes of halonium cations (e.g., [Py–I–Py]$^+$BF$_4^-$[19], [NHC–I–NHC]$^+$BF$_4^-$[20], etc.) rather than to the conventional noncovalent XB. To the best of our knowledge, the noncovalent XB interaction including the lone pair of any other classes of carbon σ-donors (isocyanides, carbon monoxide, C-ylides) has not been reported and its recognition has been challenging.

In this work, we report our discovery of a halogen bond between the lone pair of the isocyanide carbon atom and σ-holes of iodine centers of iodo-substituted perfluoroarenes. The association of isocyanide species with the XB donors considerably reduces isocyanide odor, making them more suitable for laboratory storage and usage while preserving their reactivity.

## Results

**Solution and mechanochemical routes to isocyanide adducts.** Taking into account our general interest in the isocyanide chemistry (for our relevant review see ref. [21]) and, in particular, isocyanide involving crystal engineering[22–26], we focused our efforts on isocyanide species (CNR) featuring potential lp at the C atom. As a model XB acceptor for this study we addressed the isocyanide CNMes because of its broad usage, commercial availability, and excellent solubility in nonpolar solvents. As donor components we chose four iodoperfluorobenzenes, namely iodopentafluorobenzene (abbreviated as IPFB), 1,2-diiodo-3,4,5,6-tetrafluorobenzene (1,2-FIB), 1,4-diiodo-2,3,5,6-tetrafluorobenzene (1,4-FIB), 1,3,5-triiodo-2,4,6-trifluorobenzene (1,3,5-FIB) (Fig. 1a). The colorless crystals were obtained by slow evaporation of solutions of CNMes and anyone of the iodoperfluorobenzenes in hexane at 20–25 °C. Both IPFB and 1,2-FIB form the (CNMes)•IPFB and (CNMes)•1,2-FIB adducts, whereas 1,4-FIB and 1,3,5-FIB co-crystalize with 2 equv. of CNMes to give (CNMes)$_2$•1,4-FIB and (CNMes)$_2$•1,3,5-FIB. Application of an excess of the reagents (CNMes or iodoperfluorobenzenes) or variation of solvent systems (DCM, chloroform, tetrachloromethane, 1,2-DCE, toluene, or their mixtures in various combinations were tested) do not lead to other crystalline adducts. We also obtained all four adducts by mechanochemical

liquid-assisted grinding (LAG)[9] of the two reactants in the presence of a small amount of hexane at room temperature.

The obtained adducts were characterized in the solid state by single-crystal (XRD) and powder X-ray diffraction, FTIR and solid-state $^{13}$C CP/MAS NMR spectroscopies, and also by thermogravimetric analysis (TG/DTG) (see Supplementary Methods).

**XB patterns of XRD structures of the adducts.** The plots of the XRD structures are given in Fig. 1b–e (for the crystal packing see Supplementary Figs. 3–6), and distances and angles for selected short contacts and covalent bonds are listed in Table 1.

In the structure of (CNMes)•IPFB (Fig. 1b), the iodine atom is engaged in two short I···C contacts with two isocyano groups with the interatomic distances 3.134(6) Å (yellow dotted line) and 3.831(4) Å (orange dotted line), respectively. For the former contact the atom separation is by 15% shorter than sum of Bondi vdW radii (the shortest radii known[27]; abbreviated as ΣBvdW). The latter contact is 5% longer than ΣBvdW and it is very close to the sum of Rowland vdW radii[28] (ΣRvdW). Based on these angular parameters (∠(C–I···C) 176.29(13)° and ∠(I···C≡N) 177.4(4)° are close to 180°) the first contact can be formulated as XB[8] between σh of the iodine atom and the isocyano C lp. In the second contact, ∠(C–I···C) 79.50(14)° and ∠(I···C≡N) 94.9(3)° are close to 90° and, accordingly[8], this interaction cannot be recognized as XB. Previously, we reported a similar interaction pattern for (isocyano group)···lp[23,24] and (isocyano group)···π-system[29] contacts observed for metal-bound CNR; in these systems the isocyanide fragment acted as a π-hole donor. We assume that in the case of (CNMes)•IPFB, the longer I···C contact could also have a contribution of π-hole···lp interaction involving the π-system of the isocyanide group and lp of the iodine atom.

In the XRD structure of (CNMes)•1,2-FIB (Fig. 1c), the crystallographically independent part consists of two pairs of CNMes and 1,2-FIB molecules forming two slightly asymmetrical units joined by XBs. Each 1,2-FIB provides two XB patterns, namely between one I and the isocyano C atoms (shown by yellow dotted line) and also bifurcated XB[11,12,30,31] involving the other I atom and simultaneously the isocyano C and, on the other hand, one I of the neighboring 1,2-FIB (orange dotted lines). The former XB ($d_{XB}$ = 3.029(5)/3.043(4) Å; 82% of ΣBvdW) is even shorter than that in (CNMes)•IPFB, whereas ∠(C–I···C) is also in the 170–180° range. In the case of the bifurcated contact[12,30,31], the I···C and I···I separations are close to ΣRvdW ($d$(I···C) = 3.817(5)/3.858(3) Å; $d$(I···I) = 3.9753(4)/4.0396(4) Å) and ∠(C–I···C) and ∠(C–I···I) are in the 140–170° range; these values agree well with those for other reported bifurcated XBs with $\eta^2$-(N,N)[30], -(O,O)[31], and -(Halogen,Metal)[11,12] linkages.

The XRD structure of (CNMes)$_2$•1,4-FIB (Fig. 1d) displays two symmetrical I···C contacts involving two identical isocyanide moieties ($d_{XB}$ = 3.121(3) Å; ∠(C–I···C) = 178.89(7)°; ∠(I···C≡N) = 165.35(19)°), which are almost the same as for XBs in (CNMes)•IPFB.

The XRD structure of (CNMes)$_2$•1,3,5-FIB (Fig. 1e) displays two crystallographically different types of CNMes, which, in turn, are involved in two types of XBs. One isocyanide moiety forms the linear XB (∠(C–I···C) and ∠(I···C≡N) are 180° due to the symmetry plane) and the separation is 86% of ΣBvdW. In the second CNMes, the isocyano C is engaged in two symmetrical XBs with two 1,3,5-FIB under ∠114° and grater interatomic distance, i.e., 3.334(2) Å that is 90% of ΣBvdW (Fig. 2). Involvement of one CNMes simultaneously in two XBs leads to XB-connected 1D chains, which are linked between each other by the C–H···F hydrogen bonds (HBs) to form 2D layers (Fig. 2).

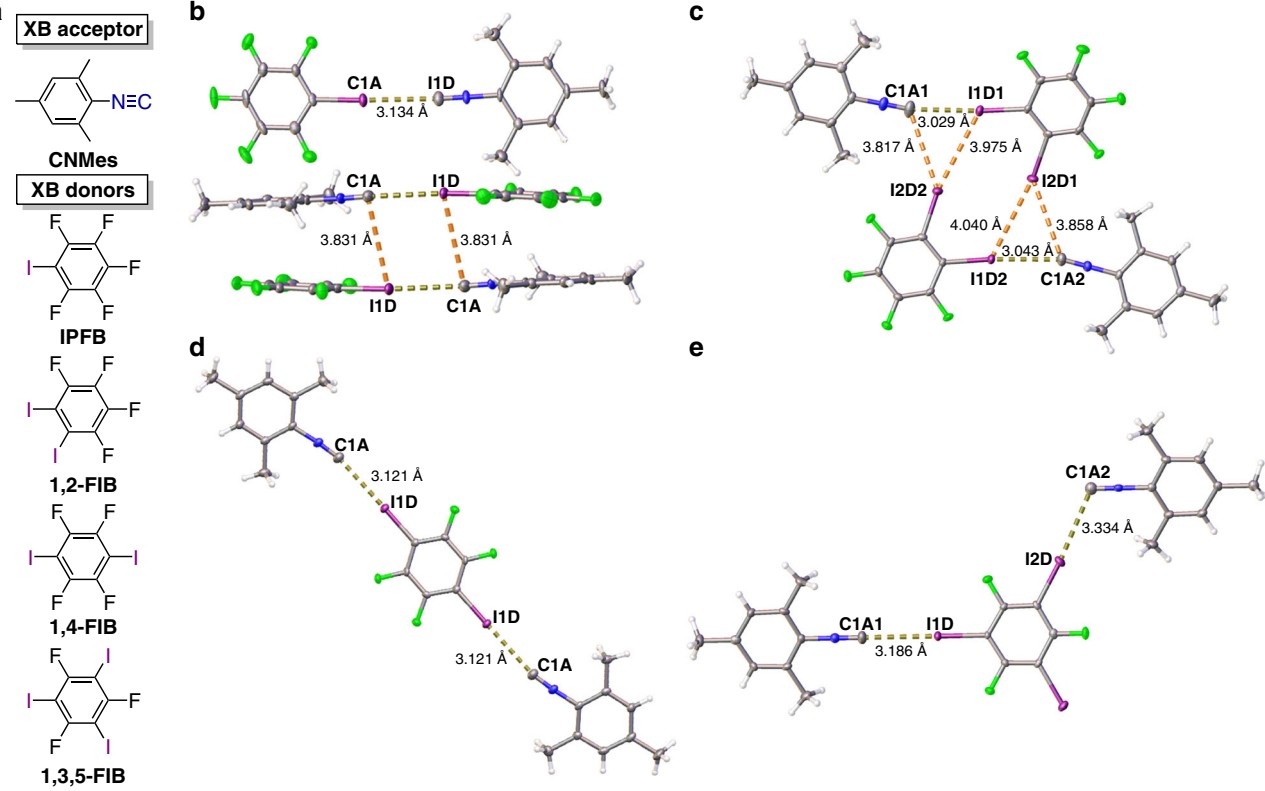

**Fig. 1 XB adducts with CNMes. a** Schematic view of CNMes and the iodoperfluorobenzenes. The XRD structures of the adducts: **b** (CNMes)•IPFB, **c** (CNMes)•1,2-FIB, **d** (CNMes)$_2$•1,4-FIB, **e** (CNMes)$_2$•1,3,5-FIB. Dotted lines indicate the identified noncovalent interactions.

**Table 1 Geometrical parameters for short contacts in the adducts.**

| Contact | $d$(I⋯X), Å | ΣBvdW, Å | $R_{XB}$(BvdW),%[a] | ΣBvdW, Å | $R_{XB}$(RvdW),%[a] | ∠(C–I⋯X), (°) | ∠(I⋯X-Y), (°) |
|---|---|---|---|---|---|---|---|
| **(CNMes)•IPFB** | | | | | | | |
| I1D⋯C1A | 3.134 (6) | 3.68 | 85 | 3.80 | 82 | 176.29 (13) | 177.4 (4) |
| C1A⋯I1D | 3.831 (4) | 3.68 | 104 | 3.80 | 101 | 79.50 (14) | 94.9 (3) |
| **(CNMes)•1,2-FIB** | | | | | | | |
| I1D1⋯C1A1 | 3.029 (5) | 3.68 | 82 | 3.80 | 80 | 173.22 (15) | 169.2 (4) |
| I2D1⋯C1A2 | 3.858 (3) | 3.68 | 105 | 3.80 | 102 | 131.45 (13) | 123.3 (3) |
| I2D1⋯I2D2 | 4.040 (1) | 3.96 | 102 | 4.06 | 99 | 151.11 (9) | 119.72 (9) |
| I1D2⋯C1A2 | 3.043 (4) | 3.68 | 82 | 3.80 | 80 | 176.04 (11) | 164.9 (3) |
| I2D2⋯C1A1 | 3.817 (5) | 3.68 | 104 | 3.80 | 101 | 140.82 (13) | 119.2 (3) |
| I2D2⋯I1D1 | 3.975 (1) | 3.96 | 100 | 3.96 | 98 | 165.40 (10) | 117.78 (9) |
| **(CNMes)$_2$•1,4-FIB** | | | | | | | |
| I1D⋯C1A | 3.121 (3) | 3.68 | 85 | 3.80 | 82 | 178.89 (7) | 165.35 (19) |
| **(CNMes)$_2$•1,3,5-FIB** | | | | | | | |
| I1D⋯C1A1 | 3.186 (5) | 3.68 | 86 | 3.80 | 83 | 180.0 | 180 |
| I2D⋯C1A2 | 3.334 (2) | 3.68 | 90 | 3.80 | 88 | 175.37 (12) | 114.39 (9) |

[a]$R_{XB}$—normalized distance parameter ([$d_{XB}$/ΣvdW] × 100%).

Summarizing, I⋯C contacts in all four adducts are significantly shorter than the sum of Bondi (82–90%) and Rowland (80–88%) vdW radii and the corresponding angles are in the range 173–180° fully consistently with the IUPAC definition of XB[1]. As the studied XBs belong to the category of noncovalent interactions, they are still significantly longer than the covalent I–C$_{sp}$ bond in iodine cyanide or iodoacetylenes (ca. 2.0 Å; 54% of ΣBvdW)[32]. The isocyanides as XB acceptors are capable of forming XB with substantial variations of ∠(I⋯C≡N), viz. 117/119 and 173/176° for two asymmetric units of (CNMes)•1,2-FIB and 114 and 180° for (CNMes)$_2$•1,3,5-FIB. This feature also could be observed in the structures of halo-substituted arylisocyanides retrieved from the Cambridge Structural Database (CSD): in 4-halophenylisocyanide, ∠(Hal⋯C≡N) is ca. 180°, whereas for 1,3,5-trihalophenylisocyanides it is around 120° (Supplementary Table 3).

The formation of XBs with the isocyano C atom leads also to a slight elongation of the covalent I–C bonds (up to 0.037 Å; higher than 3σ) in the iodoperfluorobenzenes in comparison with the parent compounds (Supplementary Table 4). These changes suggest the presence the charge transfer from the isocyano group lp to the C–I σ*-orbital.

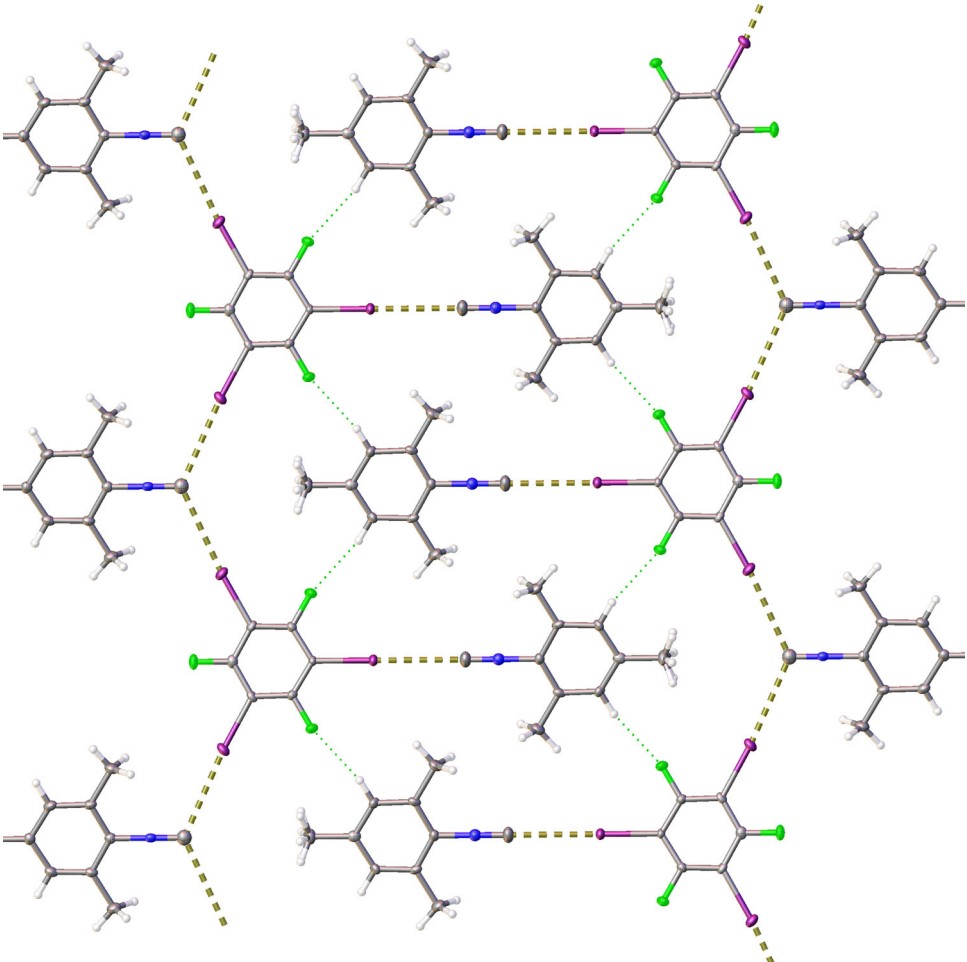

**Fig. 2 2D layer in the crystal structure of (CNMes)₂•1,3,5-FIB.** The yellow dotted lines indicate the I···C XBs, the green dotted lines depict the C–H···F HBs.

**The CSD data search for XB including carbon centers**. The processing of the CSD for halogen—(carbon lp) linkages indicates the presence of short contacts only with some isocyanides and the IAd carbene (YABJAM). According to the obtained XRD data (this work) and our processing of the CSD, such σ-donors as isocyanides and the IAd carbene under normal conditions form the shortest I···C contacts (<3.1 Å) in the crystal state. The other *C*-based XB acceptors like carbon-involved π-systems also provide short XBs ($d_{XB}$ = 3.07–3.20 Å), but mostly under high external pressure (1.9 and 3.96 GPa)[33,34]. To the best of our knowledge, XBs with other types of *C*-based σ-donors (e.g., CO or *C*-ylides), acting as XB acceptors, have never been detected. As far as XB with CNR is concerned, the CSD processing allowed the verification of only eight additional structures of CNR adducts featuring C···Halogen contacts involving the isocyano C (Supplementary Table 3). However, before our work none of these contacts was attributed to XB.

**Identification of XB by solid-state ¹³C CP/MAS NMR and FTIR**. Single-crystal XRD is the most conventional method to verify geometrical parameters of XB. NMR[35,36] and IR/Raman[37,38] spectroscopies could also be used for XB recognition in the solid state; however, these techniques are not common as they provide only collateral identification of XB by detection of spectral changes for bonded and nonbonded forms; these changes are usually less significant in the solid state than in solutions. Another complication is that the solid-state NMR for XB donor sites (Cl, Br, and I) usually gives very broad signals because of the

large quadruple moments of their NMR-active nuclei and thus it requires the application of the ultrahigh field NMR spectroscopy to obtain accurate data[35,39–41]. At the same time, the identification of XB by IR/Raman methods requires the registration of usually hard-to-reach far-IR area. On the other hand, the spectroscopic approaches for XB identification can give more information by utilization of XB acceptors featuring NMR-active nuclei (e.g., ³¹P, ⁷⁷Se)[42,43] and also those multiply bonds which are well distinguished in IR/Raman. Notably, the solid-state ¹³C NMR and FTIR have never been used for recognition of XB-involving carbon centers.

We identified XB with the isocyano C by the solid-state ¹³C CP/MAS NMR and FTIR (in KBr) spectroscopies (Table 2). The ¹³C NMR spectra for all four adducts display δ 3–6 upfield shift of the isocyano C in comparison with the parent CNMes (Supplementary Fig. 7). In (CNMes)₂•1,3,5-FIB (δ 161.06 and 164.50), the chemical shift is so sensitive that we even succeeded to distinguish two types of the isocyanide moieties, which are involved in, respectively, two types of XBs (see above). Usually the upfield shift in ¹³C NMR spectra is observed for isocyanide complexes of Lewis acids such as, for instance, transition metals[44]. Previously the relevant effect for XB acceptor was detected by solid-state ³¹P NMR for the (PPh₃)•1,3,5-FIB adduct[42].

Another evidences that prove CNR ligation to Lewis acids is the increase of ν(C≡N) frequency in IR spectra; this hypsochromic shift collaterally reflects degree of the electrophilic activation of CNR ligands upon coordination[45]. When

**Table 2 $^{13}$C NMR chemical shifts and IR $\nu(C \equiv N)$ values in the solid state for CNMes and its adducts.**

| Sample | $\delta^{13}$C, ppm | $\Delta\delta^{13}$C (free—XB), ppm | $\nu(C \equiv N)$, cm$^{-1}$ | $\Delta\nu(C \equiv N)$ (XB—free), cm$^{-1}$ | $d_{XB}$, Å |
|---|---|---|---|---|---|
| CNMes | 167.17 | – | 2116 | – | – |
| (CNMes)•IPFB | 162.82 | 4.35 | 2126 | 10 | 3.134 |
| (CNMes)•1,2-FIB | 162.95 | 4.22 | 2128 | 12 | 3.029 |
| (CNMes)$_2$•1,4-FIB | 164.15 | 3.02 | 2125 | 9 | 3.121 |
| (CNMes)$_2$•1,3,5-FIB | 164.50/ | 2.67/ | 2118 | 2 | 3.334/ |
| | 161.06 | 6.11 | | | 3.186 |

XB forms, in our experiments we observed a notable blue shift of ca. 10 cm$^{-1}$ for (CNMes)•IPFB, (CNMes)•1,2-FIB, and (CNMes)$_2$•1,4-FIB vs. the parent isocyanide. This shift is comparable to that observed upon coordination of CNR to, for instance, such metal center as Ir$^{III}$ in cyclometalated complexes[46]. Previously the relevant blue shift was observed upon XB formation with the C $\equiv$ N moiety in nitriles[47]. In (CNMes)$_2$•1,3,5-FIB exhibiting two different types of XBs, only one $\nu(C \equiv N)$ band was observed, however, with rather small frequency shifts presumably because of collective action of other noncovalent contacts such as C–H···F and π–π.

**Theoretical calculations proving the XB.** Inspection of the crystallographic data for the four adducts suggests the presence of several types of intermolecular XBs according to their geometrical parameters. In order to confirm the hypothesis on the existence of these noncovalent contacts, reveal their nature, and quantify their energies from theoretical viewpoint, we carried out DFT calculations (M06-2X/CEP-121G level of theory[48]) and performed topological analysis of the electron density distribution within the framework of Bader's theory (QTAIM method)[49] for the experimental XRD geometries of the adducts and for the optimized geometry in the gas phase for (CNMes)•IPFB associate taken as model system. Results of QTAIM analysis are summarized in Table 3 and Supplementary Table 5. The contour line diagrams of the Laplacian distribution $\nabla^2\rho(r)$, bond paths, and selected zero-flux surfaces, and reduced density gradient (RDG) isosurfaces for intermolecular noncovalent contacts in the adducts are shown in Fig. 3.

The QTAIM analysis for the experimental XRD geometries of the adducts demonstrates the presence of appropriate bond critical points (3, –1) (BCPs) between for all C···I and I···I contacts, which were verified in crystal structures according to their geometrical parameters (Table 1). The BCPs were also found for two additional I···I contacts in the structures of (CNMes)•IPFB and (CNMes)•1,2-FIB with separations that are significantly longer than $\Sigma$RvdW (Supplementary Fig. 8). Both contacts cannot be defined as true XBs[8] due to the symmetrical location of both XB donor and acceptor; the energies of these contacts are in the range 0.3–1.3 kcal·mol$^{-1}$. The low magnitude of the electron density (0.002–0.019 Hartree), positive values of the Laplacian (0.07–0.071 Hartree), and positive close to zero energy density (0.001 Hartree) in BCPs for all found contacts are typical for noncovalent interactions. The minor nonzero or zero values of the Wiberg bond indices (0.00–0.10) additionally confirms the noncovalent nature of these interactions. We have defined energies for the studied contacts according to several approaches, viz. by Espinosa et al.[50] and Vener et al.[51] approaches developed for HBs and by the Tsirelson et al. method[52] developed exclusively for noncovalent interactions involving iodine atoms (Table 3). The weakest XB with the isocyano C was observed for I···C interaction ($d_{XB}$ = 3.817(5) Å; 0.6–1.3 kcal·mol$^{-1}$) in the bifurcated contact in the structure of (CNMes)•1,2-FIB. For the direct XBs with the isocyanide their strength almost linearly

**Table 3 Energies ($E_{int}$, kcal·mol$^{-1}$) and separations ($d$, Å) for I···C and I···I noncovalent interactions in XB adducts with CNMes defined by different approaches.**

| Contact | $E_{int}{}^a$ | $E_{int}{}^b$ | $E_{int}{}^c$ | $E_{int}{}^d$ | $d_{XB}$ |
|---|---|---|---|---|---|
| (CNMes)•IPFB (XRD) | | | | | |
| I1D···C1A | 2.2 | 3.0 | 3.0 | 4.6 | 3.134 |
| C1A···I1D | 0.6 | 0.8 | 0.9 | 1.3 | 3.831 |
| I1D···I1D | 0.3 | 0.3 | 0.4 | 0.4 | 4.641 |
| (CNMes)•IPFB (gas phase) | | | | | |
| I1D···C1A | 2.8 | 3.8 | 3.8 | 5.9 | 2.978 |
| (CNMes)•1,2-FIB (XRD) | | | | | |
| I1D1···C1A1 | 2.8 | 3.5 | 3.8 | 5.5 | 3.029 |
| I2D2···C1A1 | 0.6 | 0.8 | 0.9 | 1.3 | 3.817 |
| I2D2···I1D1 | 0.9 | 1.1 | 1.3 | 1.7 | 3.975 |
| I2D1···I2D2 | 0.6 | 0.8 | 0.9 | 1.3 | 4.261 |
| (CNMes)$_2$•1,4-FIB (XRD) | | | | | |
| I1D···C1A | 2.2 | 3.0 | 3.0 | 4.6 | 3.121 |
| (CNMes)$_2$•1,3,5-FIB (XRD) | | | | | |
| I1D···C1A1 | 1.9 | 2.7 | 2.6 | 4.2 | 3.186 |
| I2D···C1A2 | 1.3 | 1.6 | 1.7 | 2.5 | 3.334 |

$^a E_{int}$ = $-V(r)/2$[50].
$^b E_{int}$ = 0.429$G(r)$[51].
$^c E_{int}$ = 0.68($-V(r)$)[52].
$^d E_{int}$ = 0.67$G(r)$[52].

depends on the atom separation in this contact and varies from 1.3–2.5 kcal·mol$^{-1}$ ($d_{XB}$ = 3.334(2) Å) in (CNMes)$_2$•1,3,5-FIB to 2.8–5.5 kcal·mol$^{-1}$ ($d_{XB}$ = 3.029(5) Å) in (CNMes)•1,2-FIB (Fig. 4).

In order to exclude the crystal-packing effects from consideration, we optimized the structure of the isolated supramolecular associate of (CNMes)•IPFB in the gas phase using the experimental XRD geometry as a starting point. The geometry optimization leads to the preservation of XB with the isocyano C in the gas phase and even noticeable shortening of the I···C contact (by 0.156 Å) making the XB separation 81% of $\Sigma$BvdW. This separation is shorter than all XBs in the experimental XRD structures of the adducts. This shortening of the I···C distance is also accompanied with the rising of XB energy up to 5.9 kcal·mol$^{-1}$ defined by QTAIM using the Tsirelson et al. approach[52]. It is noteworthy that the obtained values of the XB separation and energy still fit well to the proposed correlation between the experimental I···C distances and XB energies based on the XRD data (Fig. 4). Taking into account that sometimes interaction energies interpolated from the QTAIM analyses are considered as speculative, we also calculated the "conventional" vertical (7.6 kcal·mol$^{-1}$) and adiabatic (6.9 kcal·mol$^{-1}$) total energies for the dissociation of (CNMes)•IPFB in the gas phase (Supplementary Table 6); thus obtained energies are expectedly higher and well consistent with the results of the QTAIM analysis. Thus, both the QTAIM results and the calculated vertical/adiabatic dissociation energies reflect a rather high stability of the (CNMes)•IPFB adduct.

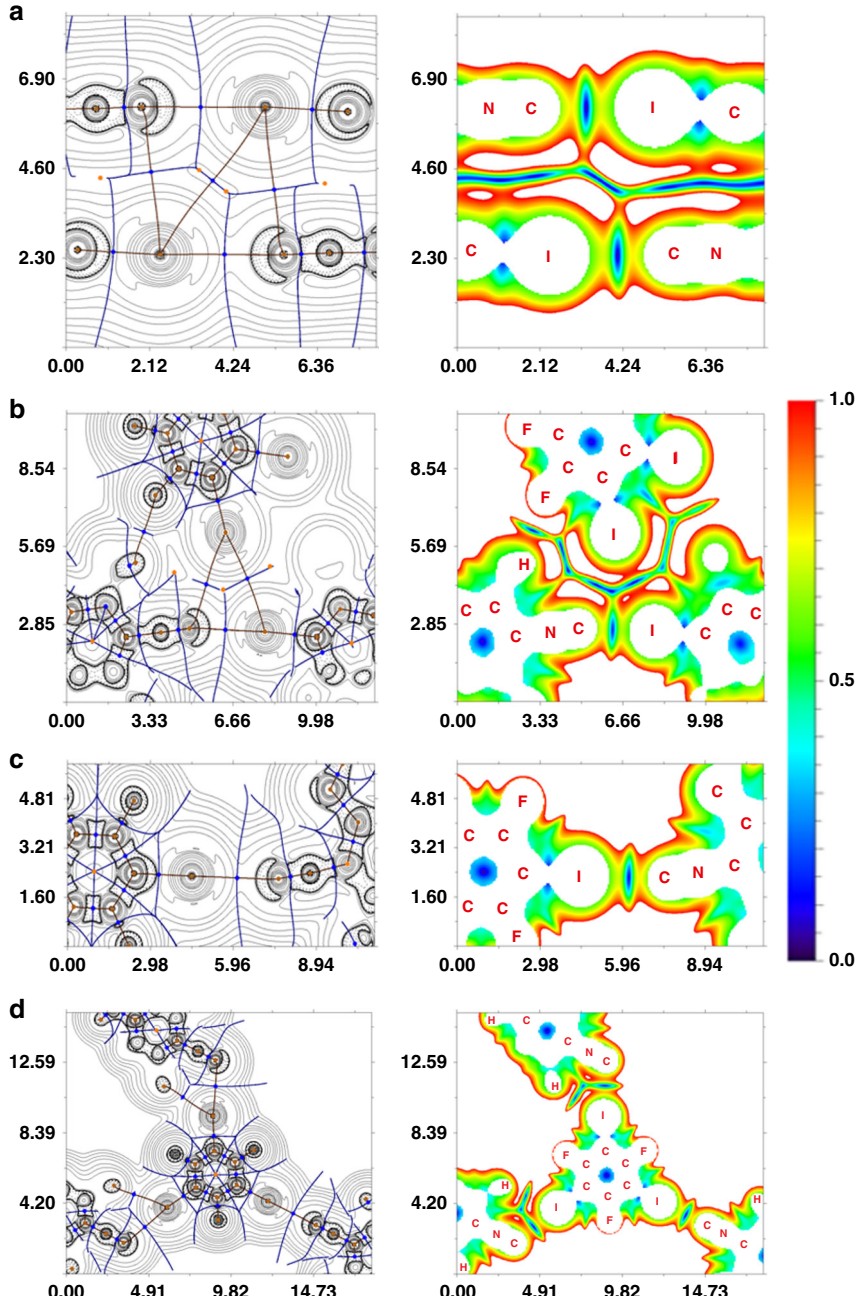

**Fig. 3 Visualization of the QTAIM analysis results for the XB adducts with CNMes.** Contour line diagrams of the Laplacian distribution $\nabla^2\rho(r)$, bond paths and selected zero-flux surfaces (left) and RDG isosurfaces (right) referring to I···C and I···I noncovalent interactions in **a** (CNMes)•IPFB, **b** (CNMes) •1,2-FIB, **c** (CNMes)$_2$•1,4-FIB, and **d** (CNMes)$_2$•1,3,5-FIB. Bond critical points (3, –1) are shown in blue, nuclear critical points (3, –3)—in pale brown, ring critical points (3, +1)—in orange. Length units—Å, RDG isosurface values are given in a.u.

The calculated IR spectra for the optimized gas phase structures of CNMes and (CNMes)•IPFB display the blue shift ($\Delta = 26\,cm^{-1}$) of $\nu(C \equiv N)$ (Supplementary Fig. 9), which is consistent with the experimental data. This shift could be caused by the presence of intermolecular charge transfer (CT) along the I···C XB in both calculated and experimental structures of (CNMes)•IPFB. To define the direction of CT in the (CNMes) •IPFB system, we applied NBO approach[53]. Second order perturbation theory analysis of the Fock matrix in NBO basis reveals two different directions of the intermolecular CT along the I···C XB (Fig. 5). The major one is CT with total E(2) value of 13.63 kcal · mol$^{-1}$ from lp of the isocyano C to the σ*(I − C)-

orbital of IPFB and this direction is typical for conventional XB. The second one has the opposite direction, i.e., from the electron belt lp of the iodine atom to σ*/π*-orbitals of isocyano group with total E(2) value of 3.64 kcal · mol$^{-1}$. This situation is not usual for conventional XBs and it is perhaps relevant to the π-back bonding, which is more typical for isocyanide ligands in complexes of electron rich metal centers[21].

The results of DFT calculations and QTAIM analysis indicate the formation of strong XB between CNMes and the iodoperfluorobenzenes; this type of XB with its relatively high energy make isocyanides promising acceptors for XB-involving crystal engineering[2]. The strong XB with isocyano group is somehow

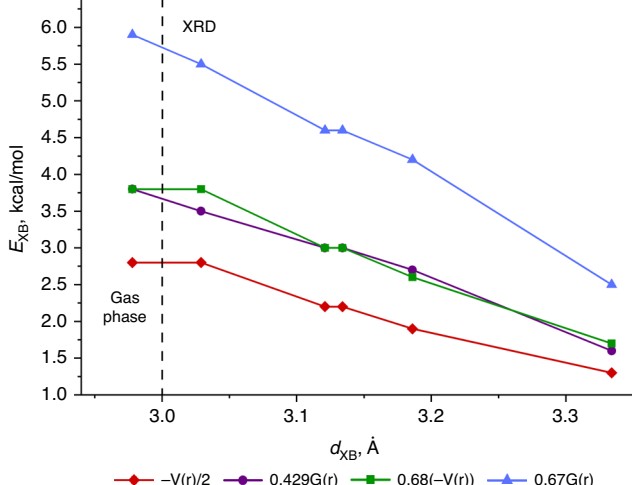

**Fig. 4 The correlation between I···C distances and XB energies in the adducts.** The correlation between I···C distances ($d_{XB}$, Å) in the experimental XRD and optimized gas phase structures of studied adducts and the XB energies (kcal · mol$^{-1}$) defined by four different approaches: $E_{XB}$ = –V(r)/2[50] (red diamonds), 0.429G(r)[51] (purple circles), 0.68(–V(r))[52] (green squares), 0.67G(r)[52] (blue triangles). Data are taken from Table 3.

$$\sigma^*/\pi^*(CN) \leftarrow n(I)$$

$$n(CN) \rightarrow \sigma^*(I-C)$$

**Fig. 5 Intermolecular charge transfer along the XB with isocyano group.** Two different directions of the intermolecular charge transfer along I···C XB in the supramolecular associate of (CNMes)•IPFB referring to the n(CN) → σ*(I − C) and n(I) → σ*/π* transitions.

of entire classes of chemicals in research practice and industrial synthesis.

Despite the high demand for isocyanide synthones in organic[60–62] and organometallic[21,63] chemistry, their use is banned in many research laboratories due to their intensely foul, penetrating odor. This specific property of volatile isocyanides is noted by many authors and well known to anyone who has ever encountered these compounds in practice, and also to their neighbors. So strong is the smell that a patent has been issued for the use of isocyanides as nonlethal chemical weapons[64].

Herein we demonstrated that the association of isocyanide species with the XB donors dramatically reduces isocyanide odor, thus defusing the stink bomb and removing arguably the main constraint on their storage and laboratory use. The odor reduction is primarily the case for adducts obtained from hexane solution; however, we also succeeded in defusing the stink bomb using samples obtained by LAG. Notably, the preparation of a mechanical mixture of mesityl isocyanide and the XB donor does not lead to the odor reduction. To achieve the desirable effect, the crystallization or liquid-assisted grinding is required.

To study quantitatively the odor reduction, we performed a series of GC-MS measurements of CNMes concentrations in the gas phase above the solid phase of the pure isocyanide and its four solid adducts (obtained via the solution approach). We observed a 3- to 46-fold decrease in the concentration of gaseous CNMes, compared to the gas phase over the same amount of parent CNMes solids (10 mg) (see Supplementary Table 7 and Supplementary Fig. 11). Although there is no clear dependence between the structures of the adducts and the degree of odor reduction, the maximum effect (46-fold) was observed for (CNMes)•1,2-FIB featuring the shortest and, consequently, the strongest XB in the series. In practice, such degree of odor reduction is sufficient for handling these compounds on an open bench without the requirement for well-ventilated hoods essential for any work involving CNRs.

Previously, a relevant approach has been used for the volatility reduction of liquid iodoperfluoroalkanes by XB-involving association with various N-heterocyclic compounds in the solid state[65] or with ionic liquids in a solution[66].

surprising taking into account quite low basicity of the isocyano C[54]. Based on the calculated values of minimum of electrostatic surface potential (ESP), the isocyanides should have comparable XB acceptor ability with the isomeric nitriles or donor substituted pyridines. At the same time, they all are inferior to aromatic N-oxides and also to the strongly donating N-heterocyclic carbenes of the push–push type (see Supplementary Discussion). Taking into account significant amount of the obtained data on Nu···I (Nu = N, O) XBs and comparable strength of these Nu···I contacts with C···I involving totally unexplored isocyanides, we believe that our findings open up a broad avenue to further exploration of XB-involving crystal engineering utilizing CNR species.

**XB-induced isocyanide odor reduction.** Olfaction plays a central role in human behavior, representing the most emotionally salient and evolutionary conserved sensory modality[55,56]. Odor associations, in particular unpleasant, enjoy a privileged brain representation[57,58], strong odors have been demonstrated to directly influence mood[55], and odor-evoked memories to disproportionately activate the hippocampus and the amygdala[59], pointing to the specific affective strength of olfactory recall. Strong aversive smells can impose decisive restrictions on the use

**Reactivity of the adducts.** The association of CNMes with iodoperfluorobenzenes dramatically reduces isocyanide odor and it makes the adducts more suitable for laboratory storage and usage, while in many instances preserving the reactivity and stability of the isocyanide. The formed XB adducts could be successfully stored in air at room temperature and exhibit close and even better thermal stability than the parent isocyanide. We compared the reactivity of the (CNMes)•IPFB adduct, taken as a model system, with the parent CNMes in some most common transformations that include ligation to metal centers and the multicomponent Ugi reaction (for details see Supplementary Methods). We found that under the same conditions both the model XB adduct and the parent isocyanide displayed the same reactivity in complexation to such metal centers as Au$^I$, Pd$^{II}$, Pt$^{II}$: the reactions with such common complex precursors as AuCl (THT), PdCl$_2$(MeCN)$_2$, and PtCl$_2$(EtCN)$_2$ in all cases led to the tetrahydrothiophene (THT) or nitrile ligands substitution to give the corresponding isocyanide complexes in quantitative yields. Similar results were obtained in the model multicomponent Ugi reaction[60] involving benzaldehyde, p-trifluoromethylaniline, and benzoic acid: the reaction with both the XB adduct and the free isocyanide gave the target bis-amide in comparable yields (ca. 80% $^1$H NMR yield with adduct vs. 85% with the parent CNMes). In all these tested reactions, we observed no difference between the using CNMes or its adducts, except that the work with the

isocyanide itself requires a ventilated hood, whereas the reactions with the adducts can be performed on an open bench.

It is noteworthy that the application of the isocyanide adducts, instead of the parent isocyanide, requires a larger mass of sample per the same amount of isocyanide to achieve the same reaction. On the other hand, once working with the isocyanide adducts, the side reactions including iodoperfluoroarenes should be taken into account. Their most common transformation include, for instance, nucleophilic substitution of fluorine or/and iodine atoms[67–69], metal-catalyzed C–C coupling[70], reduction[71], as well as metal-halogen exchange[72,73]. In addition, XB with isocyanides should electrophilically activate the isocyano group what could possibly promote the nucleophilic addition to the isocyano carbon, which is a typical reaction for some of metal-activated isocyanides[21].

To summarize, we report on the first recognition in the solid state of a noncovalent halogen bond involving the lone pair of carbon atom functioning as XB acceptor. In particular, the co-crystallization or mechanochemical liquid-assisted grinding of CNMes with iodo-substituted perfluoroarenes leads to the adducts featuring a XB between a lone pair of the isocyanide C atom and σ-holes of iodine centers of iodo-substituted perfluoroarenes. Apart from XRD, this halogen bond was additionally detected by solid-state [13]C NMR and FITR spectroscopies, and all our conclusions were supported by theoretical studies.

Compared to other sensory domains, strong odors have a disproportionate effect on the human brain, and in the case of isocyanides represent a critical hurdle to the practical application of these compounds. The co-crystallization dramatically reduces isocyanide odor (3- to 46-fold gas phase concentration decrease), removing a major restriction on the use of these versatile and reactive synthons in organic and organometallic chemistry. The application of the XB-involving adducts, instead of the parent CNR species, makes isocyanides more suitable for laboratory storage and usage while preserving their reactivity.

## Methods

**General information**. The mesityl isocyanide, iodoperfluorobenzenes, and all reagents and solvents used for syntheses and crystal growth were obtained from commercial sources and were used as received.

**Adducts synthesis and single-crystal growth**. Single-crystals were obtained by dissolution of a mixture of CNMes (30 mg, 0.2 mmol) with corresponding iodo-perfluorobenzene (IPFB 61 mg/0.2 mmol; 1,2-FIB 83 mg/0.2 mmol; 1,4-FIB 41 mg/0.1 mmol; 1,3,5-FIB 53 mg/0.1 mmol) in 3 mL of hexane and leaving the solution to cool and evaporate at room temperature for 1–2 d. The LAG synthesis of adducts with CNMes was produced by grinding of CNMes (30 mg, 0.2 mmol) with corresponding iodoperfluorobenzene (IPFB 61 mg/0.2 mmol; 1,2-FIB 83 mg/0.2 mmol; 1,4-FIB 41 mg/0.1 mmol; 1,3,5-FIB 53 mg/0.1 mmol) in an agate mortar along with 30.0 μL of hexane for 10 min.

**Single-crystal X-ray diffraction**. The crystals of adducts (CNMes)•IPFB, (CNMes)•1,2-FIB, (CNMes)$_2$•1,4-FIB, and (CNMes)$_2$•1,3,5-FIB were obtained from hexane solutions. For single-crystal XRD experiment, suitable crystals of (CNMes)•IPFB and (CNMes)$_2$•1,4-FIB was fixed on a micro mount, placed on an Agilent Technologies SuperNova diffractometer, and measured at 122 K and 200 K, by using monochromated CuKa radiation. Suitable crystals of (CNMes)•IPFB, (CNMes)•1,2-FIB, (CNMes)$_2$•1,4-FIB, and (CNMes)$_2$•1,3,5-FIB were fixed on a micro mount, placed on an Agilent Technologies Xcalibur Eos diffractometer, and measured at 100 K (except (CNMes)•IPFB, which was measured at 200 K) by using monochromated MoKa radiation. All structures were solved by direct methods by means of the SHELX program[74] incorporated in the OLEX2 program package[75]. The crystallographic data and some parameters of refinement are given in Supplementary Table 8. The H atoms were placed in calculated positions and were included in the refinement in the 'riding' model approximation, with $U_{iso}$(H) set to 1.5$U_{eq}$(C) and C–H 0.96 for CH$_3$ groups, with $U_{iso}$(H) set to 1.2$U_{eq}$(C) and C–H 0.93 Å for CH groups.

**Powder X-ray diffraction**. The X-ray powder diffraction data were obtained at room temperature using a Bruker D2 Phaser Desktop X-ray diffractometer equipped with a CuKα1 + 2 source. Data were scanned over the angular range 6–60°(2θ) with a step size of 0.02°(2θ) and scan rate 1°2θ/min.

**Solid-state NMR spectroscopy**. The solid-state NMR experiments were performed on a Bruker Avance III NMR 400 WB spectrometer operating at 9.4 T. The CP/MAS NMR spectra were acquired using a double-resonance 4 mm MAS Bruker probe at a resonance frequency of 101 MHz under 6–15 kHz MAS and the external TMS standard was used as the chemical shift reference. The CP contact time in all experiments was 3.5 μs with a delay between acquisitions of 2–5 s. More information, such as spinning speeds and the number of scans may be found in Supplementary Table 9.

**FTIR spectroscopy**. IR spectra were recorded on a Shimadzu IRAffinity-1S FTIR spectrometer $(4000 − 400 \ cm^{-1})$ in KBr pellets.

**DFT and QTAIM calculations**. The full geometry optimization of model structures and single point calculations based on the experimental XRD data have been carried out with the help of the Gaussian-09 program package[76] at the DFT level of theory using the M06-2X functional. This functional was specifically developed and parameterized for correct description of noncovalent interactions[48] and also it was validated for these purposes in several benchmark studies[77–79]. No symmetry restrictions have been applied during the geometry optimization procedure. In order to describe properly heavy iodine atoms by taking into account relativistic effects and use one family of basis sets for all atoms in model systems, the CEP-121G Stevens/Basch/Krauss ECP triple-split basis sets[80,81] were used for all atoms. The Hessian matrices were calculated analytically for the obtained optimized geometries of model structures in order to prove the location of correct minima on the potential energy surfaces (no imaginary frequencies). The electrostatic surface potentials for the optimized equilibrium geometries of model structures were plotted using the Chemcraft program (http://www.chemcraftprog.com). The Multiwfn program[82] was used for topological analysis of the electron density distribution with the help of the atoms in molecules (QTAIM) method developed by Bader[49]. The Wiberg bond indices were computed by using the Natural Bond Orbital (NBO) partitioning scheme[53]. The Cartesian atomic coordinates for model structures are presented in the Supplementary Data files 1 and 2.

**Processing of the CSD**. Processing of the Cambridge Structure Database (v 5.40) was performed using the ConQuest module (v2.0.3). The analysis for the C···Hal interactions was based on C···Hal distance, and C···Hal–R angle. The distances were restricted by ΣBvdW + 0.1 Å radii and angularity was restricted to the range 160–180° typical for XBs. In case of several contacts, the shortest contact was selected. Only structures with determined 3D and with no error were included in the search query. In addition to this, powder structures were excluded from the search. To ensure that we have only high-quality structures, R-factor—which represents the agreement between the obtained crystallographic model and the experimental diffraction data—was kept below 0.1. The application of the less restrictive criterion for the CSD search (no R-factor restriction, disorder, and errors allowed) does not lead to any new structures in case of the XB with isocyanides and gives only five additional structures for the general search on the short C–I···C contacts below 3.25 Å. However, the found contacts with the carbene and isocyanides are still among the shortest.

**GC-MS odor reduction measurements**. The crystalline samples of free CNMes and its XB adducts containing of 10 mg (0.07 mmol) of CNMes were placed in 15 mL sealed vials and thermostated at 30 °C. The GC-MS analyses were carried out on a GC-MS-QP2010 Ultra (Shimadzu) Gas Chromatograph Mass Spectrometer, equipped with an Agilent Technologies HP-5 ms column (0.25 μm, 60 m × 0.32 mm). Injection was performed in the split mode with split ratio 20 and injection volume 500 μL. The temperature of the injector was 100 °C, and the samples were analyzed using the following temperature program: 30 °C for 5 min, then 5 °C per min until 100 °C, followed by 15 min at 100 °C. The carrier gas (He) was used in the constant flow mode at 2.3 mL · min$^{-1}$ ($P = 100$ kPa). The analyses were performed in the electron impact (EI) mode (ionization energy 70 eV, source temperature 200 °C).

**Thermal analysis**. Melting points (Supplementary Table 10) were determined in capillaries with a Stuart SMP 30 apparatus. TG/DTG measurements were performed with a NETZSCH TG 209 F1 Libra thermoanalyzer. The initial weights of the samples were in the range 0.6–1.6 mg. The experiments were run in an open alumina crucible in a stream of argon at a heating rate of 10 K min$^{-1}$; the final temperature of these experiments was 500 °C.

## Data availability

All data are available from the authors upon reasonable request. The X-ray crystallographic coordinates for structures reported in this study have been deposited at the Cambridge Crystallographic Data Centre (CCDC), under deposition numbers 1957698–1957701, 1981526, 1981527. These data can be obtained free of charge from The Cambridge Crystallographic Data Centre via www.ccdc.cam.ac.uk/data_request/cif.

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

## Acknowledgements

This work was supported by the Russian Science Foundation. Experimental part was supported by project 19-13-00008, whereas the theoretical calculations were conducted as a part of the project 19-73-00001. Physicochemical studies were performed at the Center for Magnetic Resonance, Center for X-ray Diffraction Studies, Center for Chemical Analysis and Materials Research, and Thermogravimetric and Calorimetric Research Center (all belonging to Saint Petersburg State University).

## Author contributions

A.S.M. designed and performed all experiments, analyzed data, and prepared the manuscript. A.S.N. performed quantum chemical calculations and revised the manuscript. V.P.B. and V.Y.K supervised the work, provided guidance, prepared, and revised the manuscript.

## Competing interests

The authors declare no competing interests.
