## [Peer Review File · Nature Communications]

Reviewers' comments:

Reviewer #1 (Remarks to the Author):

In this interesting manuscript the authors report the existence in the solid state of a noncovalent halogen bond involving the lone pair of carbon atom acting as XB acceptor. In particular, they have obtained cocrystal CNMs with iodo-substituted perfluoroarenes leading to the adducts featuring a novel type of XBs between the lone pair of the isocyanide C atom and σ -hole of iodine of iodo-substituted perfluoroarenes. In addition to the XRD experiments, this halogen bond has been confirmed by solid-state ^{13}C NMR and FITR spectroscopies, and the conclusions were also supported by DFT calculations.

As an application, the authors have used this novel Xb to reduce the odor of isocyanides since it represents a critical hurdle to the practical application of these compounds. The authors show that cocrystallization dramatically reduces isocyanide odor (3- to 46-fold gas phase concentration decrease), thus making isocyanides more suitable for laboratory storage and usage while preserving their reactivity.

My opinion is that this manuscript is very interesting and solid piece of work. It has been competently done and it is technically correct. The research reported is hot topic and adequate for a top journal like Nat Comm. The conclusions are well supported by the results. Therefore, I recommend acceptance after some minor revision as follows:

1. The obtained associates, apart from XRD, were characterized by solid-state methods including NMR and IR. Taking into account that carbon-iodine XB is rather strong, as verified by appropriate DFT studies, could these associates also exist in solutions? I suggest to monitor (isocyanide plus XB donor) solutions by ESI mass-spectrometry.
2. The authors should justify the choice of the M06-2X functional for analyzing the X-bonds and also the basis set, that seems quite unusual for light atoms.
3. In the CSD search, to ensure that they have only high-quality structures, the authors use a tight criterion for the R-factor (below 0.1). In principle this is a good criterion. My question is that: what does happen if a looser criterion is used? can the authors comment something about that without using those structures for further analysis?. This is just to know if there are many more examples or not.

Reviewer #2 (Remarks to the Author):

The manuscript reports a very comprehensive study regarding the identification of a novel type of halogen bond involving the lone pair of the isocyanide carbon atom. The approach used judiciously combines experimental results and DFT calculations. Furthermore, an original application is presented concerning the healthier storage of isocyanide compounds. Currently, halogen bonding is a hot topic and this interesting work fits very well the Nature Communications journal. However, I have some minor comments:

1. Line 36, the XB acronym was not defined, and CNR as well.

2. Lines 52-53, this sentence is misleading and needs to be amended: the authors recognize (lines 151-153) the earlier identification of the interaction between carbene IAd and IPFB as a halogen bond (“*coordinative* XB”).
3. Lines 159-160, eventually replace “abundant” by something like “used”, and “recognition XB” by “XB recognition”.
4. Line 193, the exact level of theory must be specified in the main text as results can strongly depend on the selected DFT functionals.
5. Lines 240-242, the authors must comment on the values of calculated vertical and adiabatic dissociation energies for the (CNMe)₃•IPFB complex, versus the interaction energies interpolated from the QTAIM analyses. For instance, the usefulness of the approach developed by Espinosa et al. in the context of HBs is not demonstrated for these XB interactions.
6. Line 254, “which is commonly accompanied isocyanide ligation” is unclear, rephrase.
7. There are some typos throughout the manuscript:
 - Line 85, “Fig. 1a” must be “Fig. 1b”; line 99, “Fig. 1b” must be “Fig. 1c”; line 110, “Fig. 1c” must be “Fig. 1d”; line 113, “Fig. 1d” must be “Fig. 1e”.
 - Line 147, “push-pull” should be “push-push”.
 - Line 166, “spectroscopic” rather than “spectroscopy”.
 - Line 167, strange mix between mass numbers and ref. numbers: “³¹P,⁴⁸⁷⁷Se⁴⁹”...
 - Line 243, “high dissociation stability” where dissociation should be removed.

Reviewer #3 (Remarks to the Author):

The paper by Mikherdov et al., titled “Defusing the Stink Bomb: The I⋯C Halogen Bond Dramatically Reduces Isocyanide Odor”, details the study of a series of cocrystals featuring mesityl isocyanide and perfluorinated halogen bond donors. The crystal structures consistently reveal a new halogen bonding motif (C-I⋯C≡N-C), with the authors observing a notable decrease in the odor typically associated with isocyanide compounds. This is aligned with recent reports on exotic halogen bonds, such as the C-I⋯P/As/Sb halogen bond (Nat. Commun. 2019, 10, 61), but instead expanding on carbon as the halogen bond acceptor (C-I⋯C). The authors have carried out extensive characterization of the prepared solids: FTIR spectroscopy, ¹³C solid-state NMR spectroscopy, melting point analysis, TG/DTG, single crystal X-ray diffraction, powder X-ray diffraction. The characterization is thorough, although I have observed some discrepancies. The article was quite interesting, novel, and will be a good contribution to the field of halogen bonding.

However, in its current state, I am not convinced that the article fits in with the broad readership of Nature Communications, while it could fit in nicely in a more specialized journal on crystal engineering. I have a series of concerns and suggestions for the authors:

Concerns:

- 1) On line 52/53 of the manuscript, the authors report that “Noncovalent XB interaction including the carbon lone pair of any classes of compounds has never been observed in the past and its recognition was challenging”, but then include in the Supporting Information a crystal structure of iodoperfluorobenzene interacting with an N-heterocyclic carbene via what appears to be a C-I...C halogen bond. The authors have supported their claim to the novelty of their own C-I...C halogen bond using DFT calculations, on the grounds that the halogen bond to the carbene in the previously reported structure is stronger and has a higher “degree of covalency” and should therefore not be considered a “noncovalent” interaction. I am not entirely convinced by the DFT calculations, as the dissociation energies of the C-I...C interaction is less than half that of the C-I bond (if I understood correctly). The experimental X-ray crystal structure featuring the C-I...C interaction to the carbene appears to me to be a non-covalent halogen bond with a length of 2.754 Å (75% of vdW radii), whereas the (CNMES)₂·1,4-FIB structure has a C-I...C bond length of 3.029 Å (85% of vdW radii). If we compare the C-I...N between cyanides and heterocycles, we observe a similar trend. For instance, in structure FUYFAI on the CSD, (iodopentafluorobenzene)(DABCO), the C-I...N length is of 2.682 Å (76% of Σ vdW, using 3.53 Å as Σ vdW). In comparison, the halogen bond observed in structure EBIHEF on the CSD, (2,4,6-trimethylbenzotrile)(pentafluoroiodobenzene), has a C-I...N length of 3.092 Å (88% of Σ vdW, using 3.53 Å as Σ vdW). As a result, I believe the authors should modify their claim to the first observation of C-I...C halogen bonds to the LP of carbon and at least include the previous report with the carbene. The authors should also mention other C-I...C halogen bonds that have been observed in other systems, such as the reports involving the π -electrons in carbon-carbon triple bonds (e.g. C-I...C with acetylenes).
- 2) The odor reduction appears to be the central theme in the abstract and title, but in the article, there is only a quick mention of this along with the reactivity tests. Further, I am concerned that the GC-MS odor reduction measurements may not be aligned with standard practice. Perhaps measuring the vapor pressures of the starting material vs. cocrystals would be a more reliable method of comparing the volatility reduction observed in their cocrystals. The authors should cite some of the previous work on showing that halogen-bonded cocrystals can reduce the vapor pressure of a chemical, see: C.B. Aakeröy et al., *Chem. Commun.* 2015, 51, 2425-2428. Moreover, the authors should modify their abstract to better reflect the content of their study, such as including mentions of the characterization work that was included in their article.
- 3) As the authors are discussing the benefits of handling these cocrystals rather than pure mesityl isocyanide due to the odour, this raises additional points that could be discussed in the manuscript. CNMES has a molar mass of 161 g/mol, whereas 1,4-diiodotetrafluorobenzene has a molar mass of 402 g/mol. This means that a larger mass of cocrystal is required to achieve the same reactions. Further, these halogen bond donors are expensive, and could be problematic in some reactions. The authors have only mentioned that “nucleophilic additions to iodoperfluoroarenes should be taken into account”. Could the authors expand on the practicalities of using these cocrystals in their reactions, along with the limitations?
- 4) The experimental section on “Solid-State NMR spectroscopy” should state that the experiments were performed on the ¹³C nucleus. Further, the authors have included the Larmor frequency of ³¹P (162 MHz) rather than 100 MHz for ¹³C, based on the specified field of 9.4 T. The authors also did not specify which chemical shift reference was used throughout the study.
- 5) In Figure S14 [¹³C SSNMR spectrum of (CNMES)·IPFB], there appears to be discrepancies between the

crystallized and liquid assisted grinding sample – a new resonance at ~18 ppm, lower intensity of the shift at 121 ppm, and broadening of the shift at 162 ppm. Discrepancies are even more visible in Figure S16 [¹³C SSNMR spectrum of (CNMES)₂·1,4-FIB] – a new resonance at ~120 ppm for the crystallized sample, missing resonance at 144.8 ppm in the LAG sample, and significant differences in the 18-22 ppm region. The ¹³C SSNMR should be very similar between the LAG/crystallized samples (within experimental error). In Figure S22, the experimental PXRD of (CNMES)·IPFB shows significant differences from the simulated pattern, while there were no PXRD for the “crystallized” sample. In Figure S24, there were additional reflections in the PXRD of (CNMES)₂·1,4-FIB. Given that the ¹³C and PXRD have discrepancies, the authors should revisit these samples to ensure high quality data and/or add a discussion on these discrepancies, if it is to be published in Nature Communications. Moreover, the authors should report the PXRDs of the crystallized samples, not just the LAG samples.

6) The authors should thoroughly review the article to correct mistakes in the English. The authors should use “fluoro” rather than “fluro” in the names of their halogen bond donors.

Suggestions & minor corrections:

1) While catchy, I do not believe that “Defusing the stink bomb” is accurate to the contents presented in this work.

2) In Table 1, there are columns with “%”, denoting the normalized distance parameter. This should be defined in the caption or with a footnote to ensure clarity.

3) Page 9, lines 163-165: In regards to “solid-state NMR for XB donor sites usually gives very broad signals because of the large quadrupole moments”, the authors should be aware of the reports on using high field NMR and Nuclear Quadrupole Resonance (NQR) spectroscopy to characterize the halogen bond donor.

4) On page 5 and 6, the references to the figures are wrong (1a, 1b, ...) and should be corrected.

5) Page 6, line 114: “two crystallographically different types of CNMES”, “types of” should be removed

6) The authors sometimes confuse halogen bond (XB) with Type I and Type II halogen···halogen contacts. While Type II halogen···halogen contacts are halogen bonds, it would be best to use “halogen contact” rather than “halogen bond” when the motif is not actually an XB. For instance, Page 12, line 214, “can be attributed to Type I XB” may be misleading. The authors should clarify what they mean by type I and type II.

7) Page 2, line 36: “CNR” should be defined here, as I could not find the definition in the manuscript.

Reviewer #4 (Remarks to the Author):

In this work, a volatile isocyanide, mesityl isocyanide, is co-crystallized with four different halogen bond donors of iodoperfluoroarenes in which the arene is benzene. Most of the discussion revolves around the crystal structures, a little spectroscopy, and some theoretical work.

Some unanswered questions are lacking in the discussion. The title indicates that this is a dramatic result with the potential to improve laboratory storage of isocyanides. Is it?

Only one isocyanide was studied.

How successful is long term storage?

How stable are the iodoperfluorobenzenes?

Do they represent any environmental hazards?

Is the odor already reduced before crystals form?

Can different mole fractions be equally successful?

Can the interaction between donor and acceptor be detected by UV-Vis?

Table 1 gives geometrical parameters that are already depicted in Figure 1 (the atom labels do not aid in the discussion since the atoms are not labelled in the Figure and they are not used in the discussion). The Figure numbers on pp. 5-6 are not referenced correctly, e.g., Fig. 1*d* is actually Fig. 1*e*. The discussion is hard to follow and can be considerably shortened.

p. 6, line 110 the formula is incorrect.

There are many errors in English usage, and iodoper*fluorobenzenes* is misspelled most of the time.

How does the halogen bonding ability of iodoperfluorobenzenes compare to that of the iodoperfluoroalkanes?

A publication by Ulness, et al. (*J. Phys. Chem. A* 2009, 113, 14052) describing the halogen bonding between iodoperfluoroalkanes and pyridine should be cited.

Also, the idea of stabilizing volatile liquids through halogen bonding is not new and was reported in 2015 by Aakerøy, although it is interesting that the liquids being stabilized were the halogen bond donors themselves. *Chem. Commun.* 2015, 51, 2425.

Reviewer #1

1. The obtained associates, apart from XRD, were characterized by solid-state methods including NMR and IR. Taking into account that carbon-iodine XB is rather strong, as verified by appropriate DFT studies, could these associates also exist in solutions? I suggest to monitor (isocyanide plus XB donor) solutions by ESI mass-spectrometry.

Our answer. We performed additional high-resolution ESI-MS experiments to verify the XB with an isocyanide in solutions. The HRESI-MS⁻ spectrum of a dichloromethane solution of a mixture of CNMes and IPFB (molar ratio 1:1) displays a set of low intensity peaks corresponding to [M – H]⁻ ion of the (CNMes)•IPFB adduct (*m/z* 437.9122) thus providing some evidences indicating that the XB is preserved in the solution. However, taking into account that (i) these evidences are only collateral; (ii) the generation of the adduct could occur in mass-spectrometry capillary under the conditions of the HRESI-MS⁻ experiment; (iii) the observed peaks are of low intensity, the arguments favoring the XB in solutions are not that solid to be included in the manuscript. We thank the reviewer for this stimulating idea, and we plan to preform a comprehensive study of the XB with isocyanides in a solution as a continuation of this project.

2. The authors should justify the choice of the M06-2X functional for analyzing the X-bonds and also the basis set, that seems quite unusual for light atoms.

Our answer. In line with the reviewer's suggestion, we added appropriate comments about the choice of the DFT functional and basis sets for theoretical calculations in the revised version of our manuscript (see section "Methods" and subsection "DFT and QTAIM" subsection):

"The full geometry optimization of model structures and single point calculations based on the experimental XRD data have been carried out with the help of the Gaussian-09 program package⁷³ at the DFT level of theory using the M06-2X functional. This functional was specifically developed and parameterized for correct description of noncovalent interactions⁵⁵ and also it was validated for these purposes in several benchmark studies.⁷⁴⁻⁷⁶ No symmetry restrictions have been applied during the geometry optimization procedure. In order to describe properly heavy iodine atoms by taking into account relativistic effects and use one family of basis sets for all atoms in model systems, the CEP-121G Stevens/Basch/Krauss ECP triple-split basis sets^{77,78} were used for all atoms." (see p. 19)

3. In the CSD search, to ensure that they have only high-quality structures, the authors use a tight criterion for the R-factor (below 0.1). In principle this is a good criterion. My question is that: what does happen if a looser criterion is used? can the authors comment something about that without using those structures for further analysis? This is just to know if there are many more examples or not.

Our answer. The application of the less restrictive criterion for the CSD search (no R-factor restriction, disorder, and errors allowed) does not lead to any new structures in the case of the XB with isocyanides and it gives only 5 additional structures for the general search on the C–

I··C contacts below 3.25 Å. However, in the latter cases, the contacts with isocyanides and carbenes are still among the shortest.

The appropriate comment was added to the experimental part in the revised version of manuscript (see p. 20): “The application of the less restrictive criterion for the CSD search (no *R*-factor restriction, disorder, and errors allowed) does not lead to any new structures in case of the XB with isocyanides and gives only five additional structures for the general search on the short C–I··C contacts below 3.25 Å. However, the found contacts with the carbene and isocyanides are still among the shortest.”

Reviewer #2

1. Line 36, the XB acronym was not defined, and CNR as well.

Our answer. The appropriate corrections were done in accord with the reviewer’s suggestion. The XB and CNR terms were defined in the text at their first appearance.

2. Lines 52-53, this sentence is misleading and needs to be amended: the authors recognize (lines 151-153) the earlier identification of the interaction between carbene IAd and IPFB as a halogen bond (“*coordinative* XB”).

Our response to reviewer 2 and reviewer 3 (see later) comments. We thank both reviewers for these important remarks. Our statement on the different classifications for C–I··C XBs in the adducts of iodopentafluorobenzene (IPFB) with the carbene (IAd) and isocyanide (CNMes) species is mostly based on the comparison of their Wiberg bond indexes (WBI), which represent the degree of covalency of the interaction (0 – no covalency, 1 – single bond / *J. Comput. Chem.* **2007**, 28, 204-221). In the (CNMes)•IPFB adduct with moderate strength XB, WBI is only 0.1 and this value reflects a low degree of XB covalence. At the same time, the calculations for relatively *strong* XB in the optimized structure of IAd•IPFB result in the WBI value of 0.31. This value is closer to the WBI of the classic example of *coordinative* [N–I–N]⁺ XB in bis(pyridine)iodonium salts (WBI is 0.43 / *Aust. J. Chem.* **2013**, 66, 1179–1188) than to one of the adduct with isocyanide. Thus, we believe that the XB in the IAd•IPFB adduct belongs to the *coordinative* XBs rather than to the conventional *noncovalent* XBs.

However, we agree with the reviewer that this example of XB with the carbene should be better highlighted in the manuscript and, therefore, we modified the appropriate sentences (initially on lines 52–53) in the revised version of the manuscript and now they sound as the following:

“The interaction in the solid state involving the carbon lone pair (lp) and halogen atom has been previously observed in only one case, namely in the adduct (IAd)•IPFB formed upon interplay of iodopentafluorobenzene and *N*-heterocyclic carbene (IAd).²⁷ However, in (IAd)•IPFB, the formed C–I bond ($d_{\text{XB}} = 2.754(3)$ Å; 75% of $\sum \text{BvdW}$) exhibits a strong covalent contribution and significant charge transfer (as follows from our DFT calculations, see **Section S3.1**, the Supporting Information). Accordingly, this linkage could be attributed to the *strong*

XBs²⁸ and by nature it is closer to the so-called *coordinative* XB²⁹ as in complexes of halonium cations (e.g. [Py-I-Py]⁺BF₄⁻,³⁰ [NHC-I-NHC]⁺BF₄⁻,³¹ etc.) rather than to the conventional *noncovalent* XB. To the best of our knowledge, the *noncovalent* XB interaction including the lone pair (lp) of any other classes of carbon σ -donors (isocyanides, carbon monoxide, C-ylides) has never been reported in the past and its recognition was challenging.” (p. 3)

3. Lines 159-160, eventually replace “abundant” by something like “used”, and “recognition XB” by “XB recognition”.

Our answer. The corrections were performed in accord with the reviewer’s suggestion and we replaced “abundant” by “conventional”.

4. Line 193, the exact level of theory must be specified in the main text as results can strongly depend on the selected DFT functionals.

We specified the exact level of theory in the appropriate sentence of the revised version of manuscript: “... we carried out DFT calculations (M06-2X/CEP-121G level of theory)” ... (see p. 10)

5. Lines 240-242, the authors must comment on the values of calculated vertical and adiabatic dissociation energies for the (CNMes)•IPFB complex, versus the interaction energies interpolated from the QTAIM analyses. For instance, the usefulness of the approach developed by Espinosa et al. in the context of HBs is not demonstrated for these XB interactions.

Our answer. In accord with the reviewer’s suggestion, the appropriate sentences were modified and they sound as the following: “Taking into account that sometimes interaction energies interpolated from the QTAIM analyses are considered as speculative, we also calculated the “conventional” vertical (7.6 kcal/mol) and adiabatic (6.9 kcal/mol) total energies for the dissociation of (CNMes)•IPFB in the gas phase (**Table S6**, the Supporting Information); thus obtained energies are expectedly higher and well consistent with the results of QTAIM analyses. Thus, both the QTAIM results and the calculated vertical/adiabatic dissociation energies reflect a rather high stability of the (CNMes)•IPFB adduct.” (see p. 13)

6. Line 254, “which is commonly accompanied isocyanide ligation” is unclear, rephrase.

Our answer. The appropriate sentence in the revised version of manuscript was modified as the following: “...which is more typical for isocyanide ligands in the complexes of electron rich metal centers.”. (see p. 14)

7. There are some typo throughout the manuscript:

- Line 85, “Fig. 1a” must be “Fig. 1b”; line 99, “Fig. 1b” must be “Fig. 1c”; line 110, “Fig. 1c” must be “Fig. 1d”; line 113, “Fig. 1d” must be “Fig. 1e”.

- Line 147, “push-pull” should be “push-push”.
- Line 166, “spectroscopic” rather than “spectroscopy”.
- Line 167, strange mix between mass numbers and ref. numbers: “³¹P, ⁴⁸Se⁷⁷” ...
- Line 243, “high dissociation stability” where dissociation should be removed.

Our answer. All these corrections were conducted.

Reviewer #3

1.1) On line 52/53 of the manuscript, the authors report that “Noncovalent XB interaction including the carbon lone pair of any classes of compounds has never been observed in the past and its recognition was challenging”, but then include in the Supporting Information a crystal structure of iodoperfluorobenzene interacting with an N-heterocyclic carbene via what appears to be a C-I···C halogen bond. The authors have supported their claim to the novelty of their own C-I···C halogen bond using DFT calculations, on the grounds that the halogen bond to the carbene in the previously reported structure is stronger and has a higher “degree of covalency” and should therefore not be considered a “noncovalent” interaction. I am not entirely convinced by the DFT calculations, as the dissociation energies of the C-I···C interaction is less than half that of the C-I bond (if I understood correctly). The experimental X-ray crystal structure featuring the C-I···C interaction to the carbene appears to me to be a non-covalent halogen bond with a length of 2.754 Å (75% of vdW radii), whereas the (CNMES)₂·1,4-FIB structure has a C-I···C bond length of 3.029 Å (85% of vdW radii). If we compare the C-I···N between cyanides and heterocycles, we observe a similar trend. For instance, in structure FUYFAI on the CSD, (iodopentafluorobenzene)(DABCO), the C-I···N length is of 2.682 Å (76% of Σ vdW, using 3.53 Å as Σ vdW). In comparison, the halogen bond observed in structure EBIHEF on the CSD, (2,4,6-trimethylbenzotrile)(pentafluoriodobenzene), has a C-I···N length of 3.092 Å (88% of Σ vdW, using 3.53 Å as Σ vdW). As a result, I believe the authors should modify their claim to the first observation of C-I···C halogen bonds to the LP of carbon and at least include the previous report with the carbene.

Our response to reviewer 2 and reviewer 3 (see later) comments. We thank both reviewers for these important remarks. Our statement on the different classifications for C-I···C XBs in the adducts of iodopentafluorobenzene (IPFB) with the carbene (IAd) and isocyanide (CNMes) species is mostly based on the comparison of their Wiberg bond indexes (WBI), which represent the degree of covalency of the interaction (0 – no covalency, 1 – single bond / *J. Comput. Chem.* **2007**, 28, 204-221). In the (CNMes)·IPFB adduct with moderate strength XB, WBI is only 0.1 and this value reflects a low degree of XB covalence. At the same time, the calculations for relatively *strong* XB in the optimized structure of IAd·IPFB result in the WBI value of 0.31. This value is closer to the WBI of the classic example of *coordinative* [N-I-N]⁺ XB in bis(pyridine)iodonium salts (WBI is 0.43 / *Aust. J. Chem.* **2013**, 66, 1179–1188) than to one of the adduct with isocyanide. Thus, we believe that the XB in IAd·IPFB adduct belongs to the *coordinative* XBs than to the conventional *noncovalent* XBs.

However, we agree with the reviewer that this example of XB with carbene should be better highlighted in the manuscript and, therefore, we modified the appropriate sentences

(initially on lines 52–53) in the revised version of the manuscript and now they sound as the following:

“The interaction in the solid state involving the carbon lone pair (lp) and halogen atom has been previously observed in only one case, namely in the adduct (IAd)•IPFB formed upon interplay of iodopentafluorobenzene and *N*-heterocyclic carbene (IAd).²⁷ However, in (IAd)•IPFB, the formed C–I bond ($d_{\text{XB}} = 2.754(3) \text{ \AA}$; 75% of ΣBvdW) exhibits a strong covalent contribution and significant charge transfer (as follows from our DFT calculations, see **Section S3.1**, the Supporting Information). Accordingly, this linkage could be attributed to the *strong* XBs²⁸ and by nature it is closer to the so-called *coordinative* XB²⁹ as in complexes of halonium cations (e.g. $[\text{Py-I-Py}]^+\text{BF}_4^-$,³⁰ $[\text{NHC-I-NHC}]^+\text{BF}_4^-$,³¹ etc.) rather than to the conventional *noncovalent* XB. To the best of our knowledge, the *noncovalent* XB interaction including the lone pair (lp) of any other classes of carbon σ -donors (isocyanides, carbon monoxide, C-ylides) has never been reported in the past and its recognition was challenging.” (p. 3)

1.2) *The authors should also mention other C-I...C halogen bonds that have been observed in other systems, such as the reports involving the π -electrons in carbon-carbon triple bonds (e.g. C-I...C with acetylenes).*

Our answer. Brief comments and appropriate citation of reports on other types of C–I...C halogen bonding involving the π -electrons in carbon–carbon double, triple, and aromatic bonds are now provided in the revised version of manuscript and cited in Refs. [24–26] (p. 3).

2.1) *The odor reduction appears to be the central theme in the abstract and title, but in the article, there is only a quick mention of this along with the reactivity tests. Further, I am concerned that the GC-MS odor reduction measurements may not be aligned with standard practice. Perhaps measuring the vapor pressures of the starting material vs. cocrystals would be a more reliable method of comparing the volatility reduction observed in their cocrystals.*

Our answer. Indeed, vapor pressure can be measured by TGA using the Knudsen method (D.M. Price, *Thermochimica Acta*, **2001**, 367–368, 253–262), which allows the association of the mass loss rate of a sample with its vapor pressure. However, this method has a limited application for di- and multicomponent systems. In practice, this approach is applicable when one component of the mixture exhibit incomparable higher volatility than (an)other component(s), whose vapor pressure(s) can be neglected. In our case, unfortunately, IPFB is even more volatile than the parent CNMes, whereas 1,2-FIB has a comparable volatility with CNMes. Hence, we think our direct GC-MS method is more convenient and we suggest leaving our data as is.

2.2) *The authors should cite some of the previous work on showing that halogen-bonded cocrystals can reduce the vapor pressure of a chemical, see: C.B. Aakeröy et al., Chem. Commun. 2015, 51, 2425-2428.*

Our answer. We are grateful to the reviewer for information about this important work that we missed. We added appropriate citation (Ref. 62) and briefly commented on this work in the odor

reduction part: “Previously, a relevant approach has been used for the volatility reduction of liquid iodoperfluoroalkanes by XB-involving association with various *N*-heterocyclic compounds in the solid state⁶² or with ionic liquids in a solution.⁶³” (see p. 15)

2.2) *Moreover, the authors should modify their abstract to better reflect the content of their study, such as including mentions of the characterization work that was included in their article.*

Our answer. In the revised version of our manuscript the abstract was modified as the following: “...The co-crystallization of model mesityl isocyanide with four iodoperfluorobenzenes leads to the first series of halogen-bonded adducts with isocyanides. The obtained adducts were characterized in the solid state by single-crystal and powder X-ray diffraction, FTIR, solid-state ¹³C CP/MAS NMR spectroscopies, and also by TG/DTG. The formation of XB with isocyanide group leads to dramatical reduction of the isocyanide odor (3- to 46-fold gas phase concentration decrease).”

3.1) *As the authors are discussing the benefits of handling these cocrystals rather than pure mesityl isocyanide due to the odour, this raises additional points that could be discussed in the manuscript. CNMES has a molar mass of 161 g/mol, whereas 1,4-diiodotetrafluorobenzene has a molar mass of 402 g/mol. This means that a larger mass of cocrystal is required to achieve the same reactions.*

Our answer. The following sentence was added in the revised version of the manuscript: “It is noteworthy that the application of an isocyanide adduct, instead of the parent isocyanide, requires a larger mass of sample per the same amount of isocyanide to achieve the same reaction.” (see p. 16)

3.2) *Further, these halogen bond donors are expensive, and could be problematic in some reactions. The authors have only mentioned that “nucleophilic additions to iodoperfluoroarenes should be taken into account”. Could the authors expand on the practicalities of using these cocrystals in their reactions, along with the limitations?*

Our answer. We modified this sentence as the following: “On the other hand, once working with the isocyanide adducts, the side reactions including iodoperfluoroarenes should be taken into account. Their most common transformation include, for instance, nucleophilic substitution of fluorine or/and iodine atoms,⁶⁴⁻⁶⁶ metal-catalyzed C–C coupling,⁶⁷ reduction,⁶⁸ as well as metal-halogen exchange.^{69, 70} In addition, XB with isocyanides should electrophilically activate the isocyanide group what could possibly promote the nucleophilic addition to the isocyanide carbon, which is a typical reaction for some of metal-activated isocyanides.”⁹” (see p. 16)

4) *The experimental section on “Solid-State NMR spectroscopy” should state that the experiments were performed on the ¹³C nucleus. Further, the authors have included the Larmor frequency of ³¹P (162 MHz) rather than 100 MHz for ¹³C, based on the specified field of 9.4 T. The authors also did not specify which chemical shift reference was used throughout the study.*

Our answer. The Larmor frequency for ^{13}C NMR experiment is indeed 100 MHz and the external TMS standard was used as the chemical shift reference. These data were added to the experimental section.

5.1) In Figure S14 [^{13}C SSNMR spectrum of (CNMES)•IPFB], there appears to be discrepancies between the crystallized and liquid assisted grinding sample – a new resonance at ~18 ppm, lower intensity of the shift at 121 ppm, and broadening of the shift at 162 ppm. Discrepancies are even more visible in Figure S16 [^{13}C SSNMR spectrum of (CNMES) $_2$ •1,4-FIB] – a new resonance at ~120 ppm for the crystallized sample, missing resonance at 144.8 ppm in the LAG sample, and significant differences in the 18-22 ppm region. The ^{13}C SSNMR should be very similar between the LAG/crystallized samples (within experimental error). In Figure S22, the experimental PXRD of (CNMES)•IPFB shows significant differences from the simulated pattern, while there were no PXRD for the “crystallized” sample. In Figure S24, there were additional reflections in the PXRD of (CNMES) $_2$ •1,4-FIB. Given that the ^{13}C and PXRD have discrepancies, the authors should revisit these samples to ensure high quality data and/or add a discussion on these discrepancies, if it is to be published in Nature Communications.

Our answer. We repeated the solid-state NMR and PXRD experiments for the crystallized and LAG samples of (CNMes)•IPFB (Fig. S15 and S23) and (CNMes) $_2$ •1,4-FIB (Fig. S17 and S25).

For LAG sample of (CNMes)•IPFB adduct the newly recorded ^{13}C NMR spectrum is fully consistent with previously measured spectrum for the crystallized (CNMes)•IPFB sample. The found discrepancies in the previous NMR spectrum for the initial LAG sample possibly could be caused by the presence of some minor impurities. In the case of the PXRD data for (CNMes)•IPFB, the discrepancies with the simulated spectra are caused by different conditions of XRD and PXRD measurements as the XRD experiment was performed at 122 K, while the PXRD data were recorded at 298 K. We performed an additional single-crystal XRD experiment at 200 K (unfortunately, the crystals decomposed at 298 K during the XRD measurements) and simulated new PXRD diffractogram, which is now fully consistent with the experimental PXRD data for crystallized and LAG samples of (CNMes)•IPFB (see Fig. S23 in the Supporting Information). Surprisingly, the temperature increase from 122 to 200 K leads to the shortening (by 0.08 Å) of I•••C XB with the isocyno group in the crystal of (CNMes)•IPFB.

In the case (CNMes) $_2$ •1,4-FIB adduct, the re-acquisition of NMR spectra also helped to eliminate the discrepancies between the crystallized and LAG samples. The newly obtained PXRD for the crystallized (CNMes) $_2$ •1,4-FIB sample is fully consistent with the previously obtained PXRD data for the LAG sample.

Comparison of the PXR D data obtained for the crystallized and LAG (CNMes)₂•1,4-FIB samples.

The fitting of these data in TOPAS 4.0 program (by the Rietveld method) to the simulated PXR D data (from the single-crystal XRD experiment at 200 K) indicate that both the crystallized and LAG samples predominantly contain the (CNMes)₂•1,4-FIB phase as discussed in the manuscript. However, the diffractograms of the samples display additional low intensity reflections (i.e., at ca. 9°) from a same minor phase.

The fitting of experimental PXRD data to the simulated one from the single-crystal XRD in TOPAS 4.0 program.

Accordingly to the PXRD and ^{13}C NMR measurements, this phase does not belong to any one of the starting materials, namely to CNMes or 1,4-FIB. The solution ^1H , ^{19}F , ^{13}C NMR spectra do not indicate the presence of any noticeable impurities in the samples.

We attempted to shed the light on this additional phase, but our inspection of a number of crystallized $(\text{CNMe})_2 \cdot 1,4\text{-FIB}$ samples did not give another type crystals suitable for single-crystal XRD. We think that the additional phase could be either another polymorphic modification of the $(\text{CNMe})_2 \cdot 1,4\text{-FIB}$ adduct, or this phase could be an impurity, whose quantity is so low that it was not even detected by NMR. The appropriate comments were added in the caption to Fig. S25 in the Supporting Information.

5.3) Moreover, the authors should report the PXRDs of the crystallized samples, not just the LAG samples.

Our answer. The PXRD data of the crystallized samples were added to Fig. S23–26 in the revised version of the Supporting Information.

6) The authors should thoroughly review the article to correct mistakes in the English. The authors should use “fluoro” rather than “fluro” in the names of their halogen bond donors.

Done.

Suggestions & minor corrections:

1) *While catchy, I do not believe that “Defusing the stink bomb” is accurate to the contents presented in this work.*

Our answer. We changed the title of manuscript to the following: “The Halogen Bond with the Isocyano Carbon Provides a Dramatical Reduction of Isocyanide Odor.”

2) *In Table 1, there are columns with “%”, denoting the normalized distance parameter. This should be defined in the caption or with a footnote to ensure clarity.*

Our answer. Done.

3) *Page 9, lines 163-165: In regards to “solid-state NMR for XB donor sites usually gives very broad signals because of the large quadrupole moments”, the authors should be aware of the reports on using high field NMR and Nuclear Quadrupole Resonance (NQR) spectroscopy to characterize the halogen bond donor.*

Our answer. We modified appropriate sentence in the revised version of the manuscript: “Another complication is that the solid-state NMR for XB donor sites (Cl, Br, and I) usually gives very broad signals because of the large quadrupole moments of their NMR-active nuclei and thus it requires the application of the ultrahigh field NMR spectroscopy to obtain accurate data.^{45, 46”}

4) *On page 5 and 6, the references to the figures are wrong (1a, 1b, ...) and should be corrected.*

Our answer. Done.

5) *Page 6, line 114: “two crystallographically different types of CNMES”, “types of” should be removed*

Our answer. Done.

6) *The authors sometimes confuse halogen bond (XB) with Type I and Type II halogen···halogen contacts. While Type II halogen···halogen contacts are halogen bonds, it would be best to use “halogen contact” rather than “halogen bond” when the motif is not actually an XB. For instance, Page 12, line 214, “can be attributed to Type I XB” may be misleading. The authors should clarify what they mean by type I and type II.*

Our answer. We fully agree and the term Type I XB was omitted from the revised manuscript, while “Type II XBs” were changed on simply “XBs”. In accord with the reviewer’s suggestion, appropriate sentences (Page 12, line 214) were modified as the following: “Both contacts can not

be defined as true XBs due to the symmetrical location of both XB donor and acceptor; the energies of these contacts are in the range 0.3–1.3 kcal/mol.”

7) Page 2, line 36: “CNR” should be defined here, as I could not find the definition in the manuscript.

Our answer. Done.

Reviewer #4:

1) The title indicates that this is a dramatic result with the potential to improve laboratory storage of isocyanides. Is it?

Our results demonstrate that the formed adducts of mesityl isocyanide are noticeably less odorous than the parent isocyanide and at the same time they exhibit closer or even better stability and the same reactivity in the model reactions. Therefore, we think that co-crystallization with XB donors make isocyanides more suitable for laboratory storage and usage. Certain limitations of their applications are now indicated in the manuscript (see our response to reviewer 3 comments).

2) Only one isocyanide was studied.

Our answer. The reviewer is perfectly right and indeed we studied only one isocyanide that in our hands gives the largest amount of XRD structures. We also tested some other CNRs, but we were unable to generate a series of XRD characterized adducts.

3) How successful is long term storage?

Our answer. We performed additional powder XRD studies for some of the crystallized samples, namely for the adducts with 1,2-, 1,4-, 1,3,5-FIBs, which were obtained before the initial submission (the samples were stored in air at room temperature more than 80 days). The powder XRD data for these samples, obtained by the crystallization, agrees well with the previously obtained powder diffraction data for LAG samples and also simulated from the single-crystal XRD data. All these favor the possibility of a long-term storage. The appropriate sentence was added in the revised version of manuscript (see p. 16).

4) How stable are the iodoperfluorobenzenes?

Our answer. The solid 1,4- and 1,3,5-FIBs *reversibly* melt at 108 and 153 °C, respectively, while liquid IPFB boils at 161 °C under the normal pressure. The solid 1,2-FIB has a melting point 51 °C and it boils at 75 °C at 4 mbar pressure. Consideration of these data indicates that the perfluorobenzenes are thermally stable, at least below their melting- or boiling points. At the

same time, in accord with the obtained thermogravimetric data, the XB adducts with solid 1,2-, 1,4-, 1,3,5-FIBs exhibit similar thermal stability to the parent isocyanide and their decomposition starts in the range 110–125 °C (1% mass loss). In the adduct with liquid IPFB, the mass loss starts at a lower temperature (75 °C) and then continues with a smaller slope, what could suggest the elimination of IPFB from the crystal. The appropriate comments were added in the revised version of manuscript (see p. 16) and the Supporting Information (Section S6.4).

5) Do they represent any environmental hazards?

Our answer. All iodoperfluorobenzenes are skin, eye, and respiratory system irritants (Categories 2–physical hazard and 3–health hazard). At the same time, isocyanides belong to the category 4 (environmental hazard) acute toxicity for inhalation, oral, and dermal exposure. The formation of adducts should reduce the toxicity of isocyanide in case inhalation due to the decrease of its gas phase concentration.

6) Is the odor already reduced before crystals form?

Our answer. The preparation of a mechanical mixture of mesityl isocyanide and the XB donor does not lead to the odor reduction. To achieve the desirable effect, the crystallization or liquid-assisted grinding is required. The appropriate comments were added to the odor reduction part in the revised version of manuscript (see p. 15)

7) Can different mole fractions be equally successful?

Our answer. The formation of the XB adducts requires certain stoichiometric ratios of the starting mesityl isocyanide and iodoperfluorobenzenes and these values are given in the manuscript on p. 4. The use of other mole fractions of the starting compounds leads to the presence of residual amounts of isocyanide or iodoperfluorobenzenes in a mixture with the adduct. An excess of unbound isocyanide should increase the odor of the sample and, in the opposite situation, an excess of iodoperfluorobenzenes remains unused.

8) Can the interaction between donor and acceptor be detected by UV-Vis?

Our answer. In accord with the reviewer's suggestion, we measured UV-vis spectra for mesityl isocyanide, IPFB, and for (CNMes)•IPFB adduct in a DCM solution ($c 10^{-4} M$). Insofar as that mesityl isocyanide and IPFB absorb at the same ranges (220–250 nm), we were unable to find significant spectral changes caused by XB.

9) Table 1 gives geometrical parameters that are already depicted in Figure 1 (the atom labels do not aid in the discussion since the atoms are not labelled in the Figure and they are not used in the discussion).

Our answer. We included some of geometrical parameters of XBs both in Table 1 and Figure 1. The atom labels in Figure 1 were modified in accord with atom numeration in CIF-files and Table 1.

10) *The Figure numbers on pp. 5-6 are not referenced correctly, e.g., Fig. 1d is actually Fig. 1e. The discussion is hard to follow and can be considerably shortened. p. 6, line 110 the formula is incorrect.*

Our answer. Corrected. We moved excessive data to the Supporting Information and left only significant data.

11) *There are many errors in English usage, and iodoperfluorobenzenes is misspelled most of the time.*

Our answer. We corrected the term “iodoperfluorobenzenes” and did all our best to improve the language of the manuscript.

12) *How does the halogen bonding ability of iodoperfluorobenzenes compare to that of the iodoperfluoroalkanes?*

Our answer. According to our calculations of the maxima of the molecular surface electrostatic potential on the σ -holes on the I atoms of the optimized equilibrium model structures C_6F_5I (42.9 kcal mol⁻¹), CF_3I (46.5 kcal mol⁻¹), and $I-(CF_2)_4-I$ (45.6 kcal mol⁻¹; the M06-2X/CEP-121G level of theory) the halogen bonding ability of iodoperfluoroalkanes should be slightly higher than that of iodoperfluorobenzenes. Insofar as we did not employ iodoperfluoroalkanes in the study, we suggest leaving this part as is.

13.1) *A publication by Ulness, et al. (J. Phys. Chem. A 2009, 113, 14052) describing the halogen bonding between iodoperfluoroalkanes and pyridine should be cited.*

Our answer. Cited as Ref. [48].

13.2) *Also, the idea of stabilizing volatile liquids through halogen bonding is not new and was reported in 2015 by Aakeröy, although it is interesting that the liquids being stabilized were the halogen bond donors themselves. Chem. Commun. 2015, 51, 2425.*

Our answer. In accord with the reviewer’s suggestion, we commented on this work in the odor reduction part and cited in Ref. [62]. The appropriate sentence was added in the revised version of manuscript: “Previously, a relevant approach has been used for the volatility reduction of liquid iodoperfluoroalkanes by XB-involving association with various *N*-heterocyclic compounds in a solid state⁶² or with ionic liquids in a solution.⁶³” (see p. 15)

We hope our changes are satisfactory and the manuscript is ready for acceptance.

Best regards,

Vadim Yu. Kukushkin

(on behalf of the authors)

REVIEWERS' COMMENTS:

Reviewer #1 (Remarks to the Author):

The authors have improved the manuscript taking into consideration my previous corrections and suggestions and it is now publishable as it is.

Reviewer #2 (Remarks to the Author):

My previous comments have been satisfactorily addressed, this manuscript can be published as is.

Reviewer #3 (Remarks to the Author):

Firstly, I congratulate the authors on completing this interesting study. The authors have collected a substantial amount of data from a wide range of techniques, which clearly support the conclusions. The authors have addressed comments from all reviewers, and consequently, the overall quality of the manuscript has improved significantly. Overall, I believe that this will be a nice contribution to Nature Communications.

A minor comment that came up during my revisions:

A reviewer has suggested the authors to cite high field NMR & NQR articles:

“3) Page 9, lines 163-165: In regards to “solid-state NMR for XB donor sites usually gives very broad signals because of the large quadrupole moments”, the authors should be aware of the reports on using high field NMR and Nuclear Quadrupole Resonance (NQR) spectroscopy to characterize the halogen bond donor.”

To which the authors have cited:

45 Bryce, D. L. & Viger-Gravel, J. Solid-State NMR Study of Halogen-Bonded Adducts. *Halogen Bonding I: Impact on Materials Chemistry and Life Sciences* 358, 183-203, (2015).

46 Bryce, D. L. NMR crystallography: structure and properties of materials from solid-state nuclear magnetic resonance observables. *lucrij* 4, 350-359, (2017).

The authors should cite the primary sources rather than review articles. Here are the relevant articles to be cited:

1) F.A. Perras, D.L. Bryce. Direct investigation of covalently bound chlorine in organic compounds by solid-state ^{35}Cl NMR spectroscopy and exact spectral line-shape simulations. *Angew. Chem. Int. Ed.* 51 (2012), 4227-4230.

2) P.M.J. Szell, D.L. Bryce. ^{35}Cl solid-state NMR and computational study of chlorine halogen bond donors in single-component crystalline chloronitriles. *J. Phys. Chem. C* 120 (2016), 11121-11130.

3) P. Cerreia Vioglio, P.M.J. Szell, M.R. Chierotti, R. Gobetto, D.L. Bryce. $^{79/81}\text{Br}$ nuclear quadrupole

resonance spectroscopic characterization of halogen bonds in supramolecular assemblies. *Chem. Sci.* 9 (2018), 4555-4561.

4) P.M.J. Szell, L. Grébert, D.L. Bryce. Rapid identification of halogen bonds in co-crystalline powders via ^{127}I nuclear quadrupole resonance spectroscopy. *Angew. Chem. Int. Ed.* 58 (2019), 13479-13485.

Reviewer #4 (Remarks to the Author):

This is a thoughtful and useful report of interest to those who are concerned with the structure and properties of chemicals. The revised version is acceptable for publication.

REVIEWERS' COMMENTS:

Reviewer #1

The authors have improved the manuscript taking into consideration my previous corrections and suggestions and it is now publishable as it is.

Our answer. We are grateful to the reviewer for valuable comments, which helped up to improve this manuscript.

Reviewer #2

My previous comments have been satisfactorily addressed, this manuscript can be published as is.

Our answer. We are grateful to the reviewer for valuable comments, which helped up to improve this manuscript.

Reviewer #3

Firstly, I congratulate the authors on completing this interesting study. The authors have collected a substantial amount of data from a wide range of techniques, which clearly support the conclusions. The authors have addressed comments from all reviewers, and consequently, the overall quality of the manuscript has improved significantly. Overall, I believe that this will be a nice contribution to Nature Communications.

A minor comment that came up during my revisions:

A reviewer has suggested the authors to cite high field NMR & NQR articles. The authors should cite the primary sources rather than review articles. Here are the relevant articles to be cited.

Our answer. We incorporated citation of the suggested works, see Refs. 39–41. We are grateful to the reviewer for valuable comments, which helped up to improve this manuscript.

Reviewer #4:

This is a thoughtful and useful report of interest to those who are concerned with the structure and properties of chemicals. The revised version is acceptable for publication.

Our answer. We are grateful to the reviewer for valuable comments, which helped up to improve this manuscript.